# Natural plant products as potential inhibitors of RNA dependent RNA polymerase of Severe Acute Respiratory Syndrome Coronavirus-2

Shruti Koulgi[☯], Vinod Jani[☯], Mallikarjunachari Uppuladinne V. N.[☯], Uddhavesh Sonavane, Rajendra Joshi[iD]*

High Performance Computing—Medical and Bioinformatics Applications Group, Centre for Development of Advanced Computing (C-DAC), Pune, India

☯ These authors contributed equally to this work.
* rajendra@cdac.in

**Data Availability Statement:** All relevant data are within the manuscript and its Supporting Information files.

## Abstract

Drug repurposing studies targeting inhibition of RNA dependent RNA polymerase (RdRP) of Severe Acute Respiratory Syndrome Coronavirus 2 (SARS-CoV-2) have exhibited the potential effect of small molecules. In the present work a detailed interaction study between the phytochemicals from Indian medicinal plants and the RdRP of SARS-CoV-2 has been performed. The top four phytochemicals obtained through molecular docking were, swertia-puniside, cordifolide A, sitoindoside IX, and amarogentin belonging to *Swertia chirayita*, *Tinospora cordifolia* and *Withania somnifera*. These ligands bound to the RdRP were further studied using molecular dynamics simulations. The principal component analysis of these systems showed significant conformational changes in the finger and thumb subdomain of the RdRP. Hydrogen bonding, salt-bridge and water mediated interactions supported by MM-GBSA free energy of binding revealed strong binding of cordifolide A and sitoindoside IX to RdRP. The ligand-interacting residues belonged to either of the seven conserved motifs of the RdRP. These residues were polar and charged amino acids, namely, ARG 553, ARG 555, ASP 618, ASP 760, ASP 761, GLU 811, and SER 814. The glycosidic moieties of the phytochemicals were observed to form favourable interactions with these residues. Hence, these phytochemicals may hold the potential to act as RdRP inhibitors owing to their stability in binding to the druggable site.

## Introduction

The Severe Acute Respiratory Syndrome Coronavirus 2 (SARS-CoV-2) has led to the current pandemic across the globe, leading to a large number of fatalities [1]. This pandemic is still ongoing and significant rise in the number of patients infected worldwide has been observed day by day. There is an immediate need to find novel preventive and therapeutic agents to combat the effect of this virus. The SARS-CoV-2, has RNA genome of around 30 K nucleotides. This genome codes for the structural, non-structural (nsp) and accessory proteins, required for the viral assembly, replication and poly-protein functioning [2–4]. Therefore,

**Funding:** This work was funded by the Ministry of Electronics and Information Technology, Government of India, under the project, National Supercomputing Mission (NSM). The funders had no role in study design, data collection and analysis, decision to publish, or preparation of the manuscript.

**Competing interests:** The authors have declared that no competing interests exist.

experimental and computational approaches towards COVID-19 therapeutics, specifically targeting these proteins have gained importance in the drug industry [5]. Researchers are trying different strategies including, testing of broad-spectrum anti-viral drugs, *in-silico* screening of molecular databases and rational drug design [6, 7]. These approaches have enabled identification of lead compounds against the viral proteins based on the genomic information and pathological characteristics of COVID-19 [6]. Drug repurposing is one such strategy that is also being extensively used worldwide to design drugs against this coronavirus [7–10]. To understand how similar is the SARS-CoV-2 in comparison to the earlier known coronaviruses, several sequence comparison studies have been performed [11–17]. The phylogenetic analysis have revealed the conservation of the sequence of the coronaviruses across different species [14–16]. Amongst the various potential drug targets of SARS-CoV-2, RdRP, is known to be the most conserved among the viruses [18, 19]. RdRP has been extensively studied using various drug discovery techniques for COVID-19 therapeutics [20, 21]. A range of FDA-approved RdRP inhibitors (nucleotide analogues) for previously known viral infections have been repurposed to understand their role in inihibiting SARS-CoV-2 [22–28]. Several other FDA-approved drugs which are not, nucleotide analogues have also been screened against RdRP through *in-silico* approaches [29, 30]. These studies target towards understanding the residues involved in binding to the proposed inhibitor molecules and the mechanism of RdRP inhibition.

The three dimensional structure of the RdRP of SARS-CoV-2, resembles to that of a right cupped hand, consisting of three subdomains finger, palm and thumb [31]. The different subdomain of the RdRP of SARS-CoV-2 have been depicted in Fig 1A. The residue range of these subdomains have been shown in the Table 1. The N-terminal region of RdRP is proceeded by a nidovirus RdRp-associated nucleotidyltransferase (NiRAN) subdomain. The NiRAN and the FD subdomains are connected through the linker region. Fig 1B depicts the seven conserved motifs in the RdRP of SARS-CoV-2 and the residue range spanned by each of them. These conserved motifs flank the catalytic active site of the RdRP and hence are involved in RNA template, primer, nucleotide and inhibitor binding. The residues belonging to these functionally important sites of the RdRP have been listed in Table 1.

The available structural data of the RdRP from different viruses, was extensively analysed by Zhou Z and Bourne PE [30]. This work reported inhibitor molecules that do not belong to the class of nucleotide analogue [30]. Such studies open an avenue to explore molecules other than the nucleotide analogues as probable RdRP inhibitors. One such paradigm is the application of natural plant products as potential inhibitors against the drug target under-consideration. The glorious history of traditional medicine, involves the use of purified plant compounds in treating metabolic disorders like, diabetes and life threatening diseases like, cancer [32–34]. Similar approaches have been made in order to find a plant-based compound with a potency to treat viral infections [35–42]. Few of the detailed studies on use of natural compounds as inhibitors against coronaviruses drug targets, suggested that polyphenols, limonoids, and tri-terpenoids may be promising candidates [37, 38]. Earlier known *in-silico* and experimental studies involving phytochemicals from Indian medicinal plants, namely, *Withania Somnifera*, *Tinospora cordifolia*, *Ocimum sanctum* and *Tinospora crispa* were explored for their binding to the 3CL-protease of SARS-CoV-2 [40, 41]. Phytochemicals such as, flavonoids and structurally similar indole chalcones derivatives were investigated for their pharmokinetics and binding properties against three of SARS-CoV-2 drug targets namely, RdRP, 3CL-proteases and S-glycoprotein [42]. The availability of multiple experimental structures for the RdRP of SARS-CoV-2 along with molecular docking and simulations would enable the designing of inhibitors based on screening of phytochemicals. The present study deals with molecular docking and simulation studies of phytochemicals derived from Indian medicinal plants, with

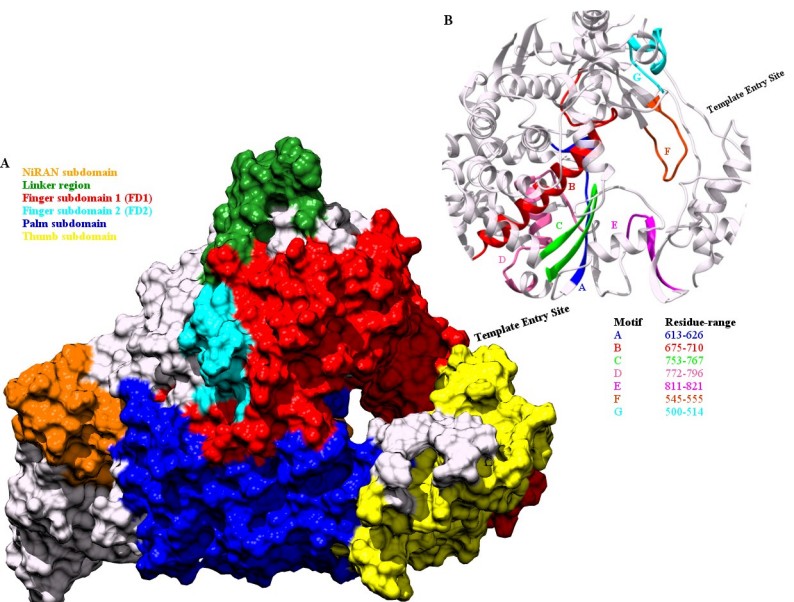

**Fig 1. Structure of RdRP of SARS-CoV-2.** The different (A) subdomains and (B) the seven conserved motifs in RdRP of SARS-CoV-2 represented through color codes.

known application in treating respiratory disorders (S1 Fig in S1 File). The comparative analysis of ligand-free and phytochemical-bound RdRP was performed based on principal component analysis (PCA), molecular mechanics-generalized Born-surface area (MM-GBSA) free energy, and hydrophilic and hydrophobic contact analysis.

## Methodology

### Model generation

The co-ordinates for the RdRP were obtained from the SWISS-MODEL, where the template used was the cryo-EM structure with PDB ID 7BTF [43]. The model consists of 932 amino acids and two $Zn^{+2}$ ions. Each $Zn^{+2}$ ion is present in a tetrahedral co-ordination complex with three CYS and one HIS residues. CYS 301, 306 and 310 along with HIS 295 form the co-ordination complex with one of the $Zn^{+2}$ atoms. CYS 487, 645, 646 and HIS 642 form the co-ordination complex with the other $Zn^{+2}$ atom.

**Table 1. Residue range and number for the subdomains and functionally important sites of the RdRP.**

| RdRP subdomains | Residue Range |
|---|---|
| Finger 1 (FD1) | 398–581 |
| Finger 2 (FD2) | 621–679 |
| Palm (PD) | 582–627 / 688–815 |
| Thumb (TD) | 816–919 |
| **Functionally important sites** | **Residue name and number** |
| Catalytic site | TRP 617, CYS 622, SER 759, ASP 760, ASP 761, CYS 813 |
| Primer binding | SER 759, ASP 760, ASP 761, CYS 813, SER 814, GLN 815 |
| Template binding | LYS 500, SER 501, ASN 507 |
| Inhibitor binding | LYS 551, ARG 553, ARG 555 |

## Phytochemical dataset

The dataset of phytochemicals was built using the protocol shown in the S1 Fig in S1 File. Initially, the information on Indian medicinal plants known to be useful in treating respiratory ailments was gathered. A total of twelve medicinal plants were selected for this study (S1 Fig in S1 File) The literature citing the medicinal use of the plants selected has been given with the S1 Fig in S1 File. The chemical structure for these phytochemicals were extracted from the Pub-Chem and CAS databases [44, 45]. The chemical structures if obtained in any format other than three dimensional MOL2 format were subjected to conversion using OpenBabel [46]. This dataset of 150 phytochemicals was further used for molecular docking studies.

## Molecular docking

The molecular docking of the phytochemical dataset was performed on the model generated for RdRP. The parameter generation was done using UCSF Chimera [47]. The AMBER14SB and generalized AMBER force field parameters were used for the RdRP and phytochemical dataset, respectively [48, 49]. The molecular docking was performed using DOCK 6 [50]. The protocol of flexible docking as reported in one of our earlier works on RdRP was followed [51]. The ligands were sorted based on the grid score, which is a measure of the effective binding of the ligand with the active site of the receptor molecule [50].

## Molecular dynamics

The molecular dynamics simulations for top four phytochemicals (based on the grid score) obtained through molecular docking were performed. Table 2 shows the grid scores for the top four phytochemicals. These phytochemicals were, swertiapuniside (SWE), cordifolide A (COR), sitoindoside IX (SIT), and amarogentin (AMR) (S2 Fig in S1 File). SWE and AMR belonged to the plant *Swertia chirayita*, COR to *Tinospora cordifolia*, and SIT to *Withania Somnifera*. The ADMET and drug likeliness properties of these phytochemicals were calculated using two servers, admetSAR 2.0 and SwissADME [52, 53]. A detailed table and a short paragraph explaining the different parameters of absorption, distribution, metabolism and toxicity of the four phytochemicals has been given in S1 Table in S1 File. The docked complexes of these phytochemicals with RdRP were used as the start structure for the ligand-bound simulations. These ligand-bound RdRP systems have been referred as RdRP-SWE, RdRP-AMR, RdRP-SIT and RdRP-COR denoting the presence of these respective phytochemicals (Table 3). The simulations were performed using the AMBER16 simulation package [54]. The force field parameters were generated using the AMBER14SB force field for the RdRP molecule [48]. The Zinc AMBER Force Field (ZAFF) was used for parameter generation of the two $Zn^{+2}$ co-ordination complexes [55]. The parameters for the phytochemicals were generated using the *antechamber* module of AMBERTOOLS17 and the force field used was the general atom force field [49, 56]. The docked complex was neutralized using Na+ ions followed by addition of solvent molecules. The octahedral geometry for TIP3P water model was used in order to solvate the ligand-bound

**Table 2. Grid scores for the top-ranked four phytochemicals obtained through DOCK 6.**

| Rank | Phytochemical Name | Grid Score (kcal/mole) |
|------|--------------------|------------------------|
| 1 | Swertiapuniside (PubChem CID:5487497) | -59.42 |
| 2 | Cordiofolide A (PubChem CID: 102451916) | -55.14 |
| 3 | Sitoindoside IX (PubChem CID: 189586) | -53.92 |
| 4 | Amarogentin (PubChem CID: 115149) | -53.39 |

**Table 3. Information about the simulations and abbreviation used.**

| Simulation System | Simulation length | Abbreviation used |
|---|---|---|
| RdRP without any ligand | 50 ns (5 replicates) ≈ 50 x 5 = 250 ns | RdRP-APO |
| RdRP + Swertiapuniside | 50 ns (2 replicates) ≈ 50 x 2 = 100 ns | RdRP-SWE |
| RdRP + Amarogentin | 50 ns (2 replicates) ≈ 50 x 2 = 100 ns | RdRP-AMR |
| RdRP + Sitoindoside IX | 50 ns (2 replicates) ≈ 50 x 2 = 100 ns | RdRP-SIT |
| RdRP + Cordifolide A | 50 ns (2 replicates) ≈ 50 x 2 = 100 ns | RdRP-COR |
| **Ligand name** | **Abbreviation used in manuscript** | |
| Swertiapuniside | SWE | |
| Amarogentin | AMR | |
| Sitoindoside IX | SIT | |
| Cordifolide A | COR | |

RdRP systems. The molecular dynamics simulations were performed using the classical steps of minimization, temperature ramping, equilibration and production run. The minimization was performed using steepest descent followed by conjugate gradient method for a cumulative of 10000 steps. Initially, the solvent was minimized followed by the solute. The entire simulation system was gradually heated up to 300 K using the Brendensen thermostat and Langevin dynamics. The hydrogen constraints were treated using the SHAKE algorithm. After achieving the desired temperature, an equilibration was performed at constant temperature of 300 K and a constant pressure of 1 atm for 2 ns. This was followed by the production run of 50 ns. Two replicates of 50 ns were simulated. This protocol was followed for all the four ligand-bound RdRP systems. Hence, for each of the four ligand-bound RdRP systems a cumulative of 100 ns simulation data was achieved. The simulation data for the ligand-free RdRP (RdRP-APO) system were obtained from the previously reported work by the authors of this article [51]. A cumulative of 250 ns simulation data for the RdRP-APO system was used for comparative study against the RdRP systems bound to phytochemicals.

## Analysis performed

The principal component analysis (PCA) and water mediated interaction analysis between the ligands and RdRP was performed using the *cpptraj* module of AMBERTOOLS17 [57]. The fluctuations were visualized using the Normal Mode Wizard (NMWiz) plugin of Visual Molecular Dynamics (VMD) [58, 59]. The *GetContacts* module of FlarePlot was used for calculating and visualizing the hydrogen bonding and salt bridge interactions between the ligand and RdRP molecules [60]. The MMPBSA.py module of AMBERTOOLS17 was used for calculating the free energy of binding between the ligand molecules and the RdRP [61]. The equation given below was used for calculating the free energy of binding ($\Delta\Delta G_{bind}$) between the RdRP protein and the bound phytochemical,

$$\Delta\Delta G_{bind} = \Delta G_{complex} - (\Delta G_{receptor} + \Delta G_{ligand})$$

$$\Delta G_{complex/receptor/ligand} = \Delta H_{complex/receptor/ligand} - T\Delta S_{complex/receptor/ligand}$$

where, $\Delta G_{complex}$, $\Delta G_{receptor}$, and $\Delta G_{ligand}$ stands for the free energy of the RdRP-Phytochemical complex, the RdRP protein and the bound phytochemical respectively. The enthalpy component, $\Delta H_{complex/receptor/ligand}$, of the free energy was considered for the calculations.

## Results and discussion

### Dominant subdomain level motions of RdRP

The conformational dynamics undergone by RdRP in the absence and presence of ligand was estimated by performing PCA on the non-hydrogen atoms of the RdRP residues for all the simulated systems. PCA enables the understanding of the dominant motions that lead to the conformational variation in the protein. Fig 2A–2D shows the distribution of RdRP conformers along PC 1 (black), PC 2 (red), and PC 3 (green) for RdRP-SWE, RdRP-AMR, RdRP-SIT and RdRP-COR respectively. The dotted line in Fig 2A represents the distribution of conformers for the RdRP-APO system. The PC 1 distribution for all the phytochemicals-bound RdRP systems showed the presence of two populations in comparison to the RdRP-APO. This infers that in the presence of phytochemicals the RdRP tends to explore different conformational zones. However, RdRP-APO system was observed to have a single dominant population. The distribution along PC 2 showed the occurrence of more than one population for the RdRP-APO (Fig 2A, dotted line), RdRP-SWE (Fig 2A) and RdRP-AMR (Fig 2B) systems. The RdRP-SIT and RdRP-COR systems showed a single population along PC 2. The basis of PCA suggests decrease in variance with increase in number of principal components. Hence,

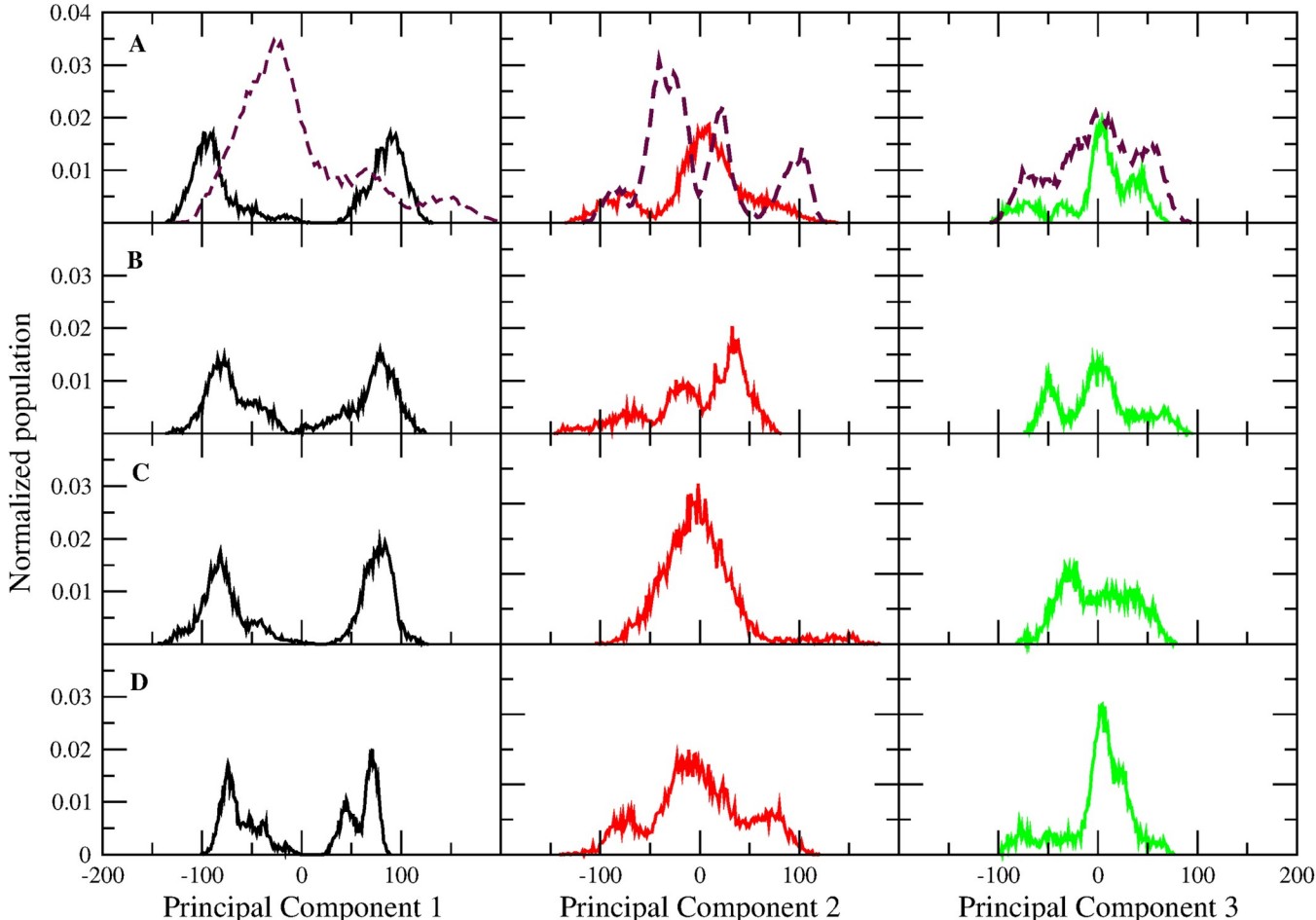

**Fig 2. PCA distribution of RdRP conformers.** Normalized population of conformers along principal component 1 (black), 2 (red), and 3 (green) for (A) RdRP-SWE, (B) RdRP-AMR, (C) RdRP-SIT, and (D) RdRP-COR systems. The dotted line in sub figure A is for the RdRP-APO system.

observing more than one population in PC 2 suggests that the conformations captured under this distribution may have occurred for a short duration within the simulation. Similarly for PC 3, multiple populations were observed but their size was comparatively lower than the first two PCs. The root mean square fluctuation (RMSF) was calculated for every residue along the first three principal components and has been given in S3 Fig in S1 File. RdRP-SWE and RdRP-SIT systems showed fluctuation in the N-terminal region which was captured by PC1. It was observed that the fluctuations in the FD1 and TD subdomain were quite significant in each of the phytochemical-bound system. To further understand what were these conformations sampled by the first three PCs the distance between FD1 and thumb subdomain were calculated. The reason behind selecting this parameter was one of the previously reported studies by Appleby and co-workers [62]. This study suggested the opening and closing of the template entry site flanked by these two domains [62].

Fig 3A–3C shows the distance between the FD1 and TD subdomain captured by PC 1, 2 and 3 for all the RdRP systems, respectively. The variation in PCs throughout the simulation was captured within 51 snapshots by the *cpptraj* module of AMBERTOOLS17. Hence, the X-axis shows maximum 51 number of frames/snapshots. The dotted line represents the distance

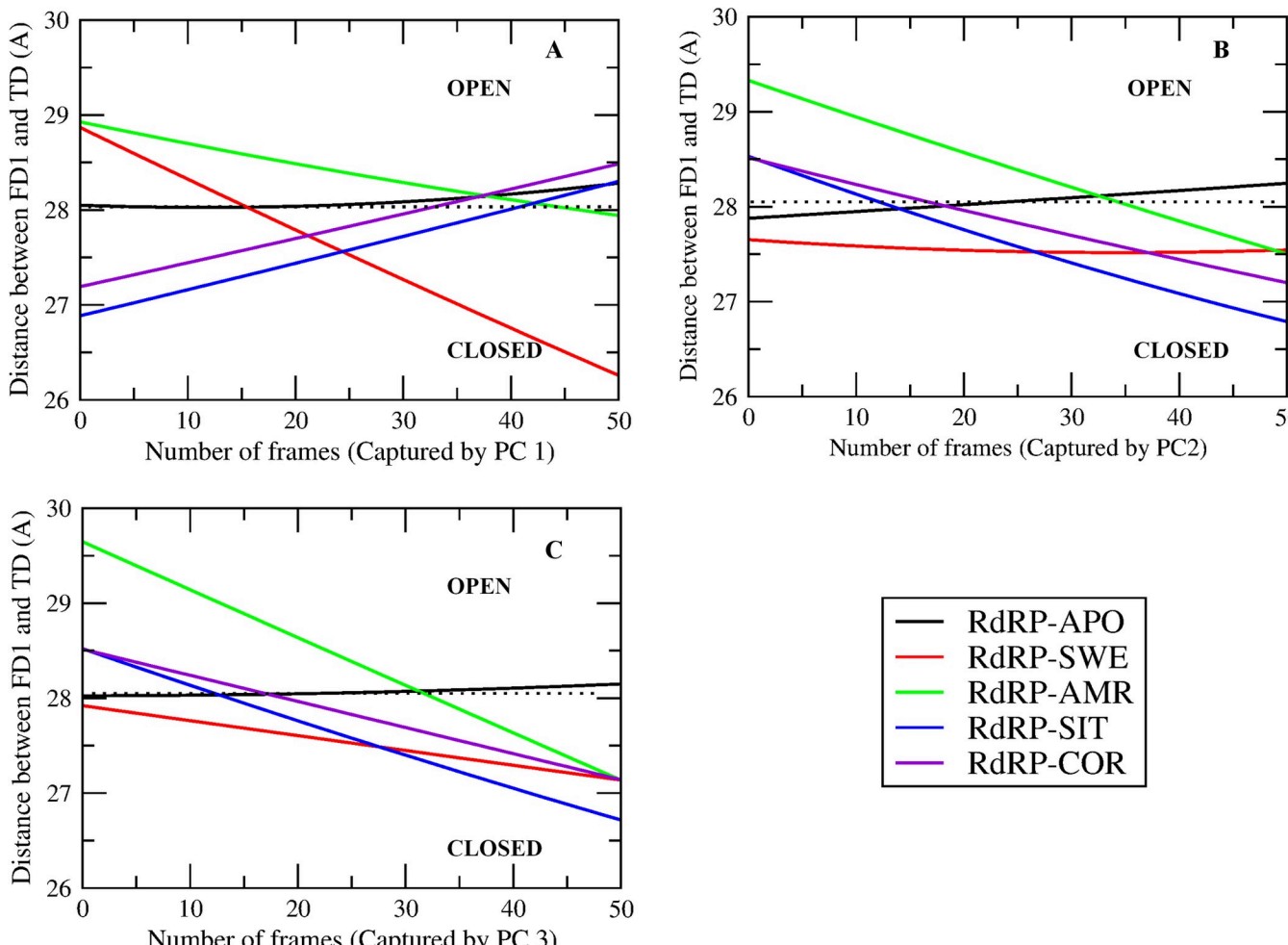

**Fig 3. Open and closed conformations of RdRP.** Distance between the finger subdomain 1 and thumb subdomain captured by principal component (A) 1, (B) 2 and (C) 3 in case of the five RdRP systems.

(28.12 Å) between FD1 and TD subdomains for the experimental structure of the apo RdRP of SARS-CoV-2 (PDB ID 7BV1) [31]. These distances greater and less than 28.12 Å suggests open and closed conformation of the template entry site, respectively. RdRP-APO (black) system was observed to have this distance close to the one seen in the experimental structure 7BV1 in the first three PCs. RdRP-SWE (red) system showed the transition from open to closed conformation of the template entry site in case of PC1 and PC3. However, in case of PC2 it was observed to be stable in the closed conformation. RdRP-AMR (green) system showed the transition from open to closed conformation in case of the first three PCs. RdRP-SIT and RdRP-COR showed the transition from closed to open conformation in case of PC1. However, the next two PCs witnessed open to closed transition similar to that observed for RdRP-SWE and RdRP-AMR. These structural changes in RdRP significantly occurred in the presence of SWE, AMR, SIT and COR. Similar observations were reported by Moustafa et al. and Thompson et al., wherein the role of finger and thumb subdomains was highlighted [63, 64]. Thus, it may be inferred that the structural changes in these two subdomains may be resultant of the binding of these phytochemicals. Hence, the molecular interactions between the phytochemicals and the RdRP residues were further investigated.

## Molecular interactions between phytochemicals and RdRP

The molecular interactions between the phytochemicals and RdRP residues were quantified by calculating the number of hydrogen bonds, salt bridges and water-mediated bonds formed between the two (Fig 4). Fig 4 shows the contact frequency of the interactions formed by the ligand. The interactions with contact frequency more than 0.2 have been plotted. The X-axis mentions the name of the residues with the motif/subdomain they belong to in parentheses. The average hydrogen bond distance between these residues and atoms of the ligands have been given in S2 Table in S1 File.

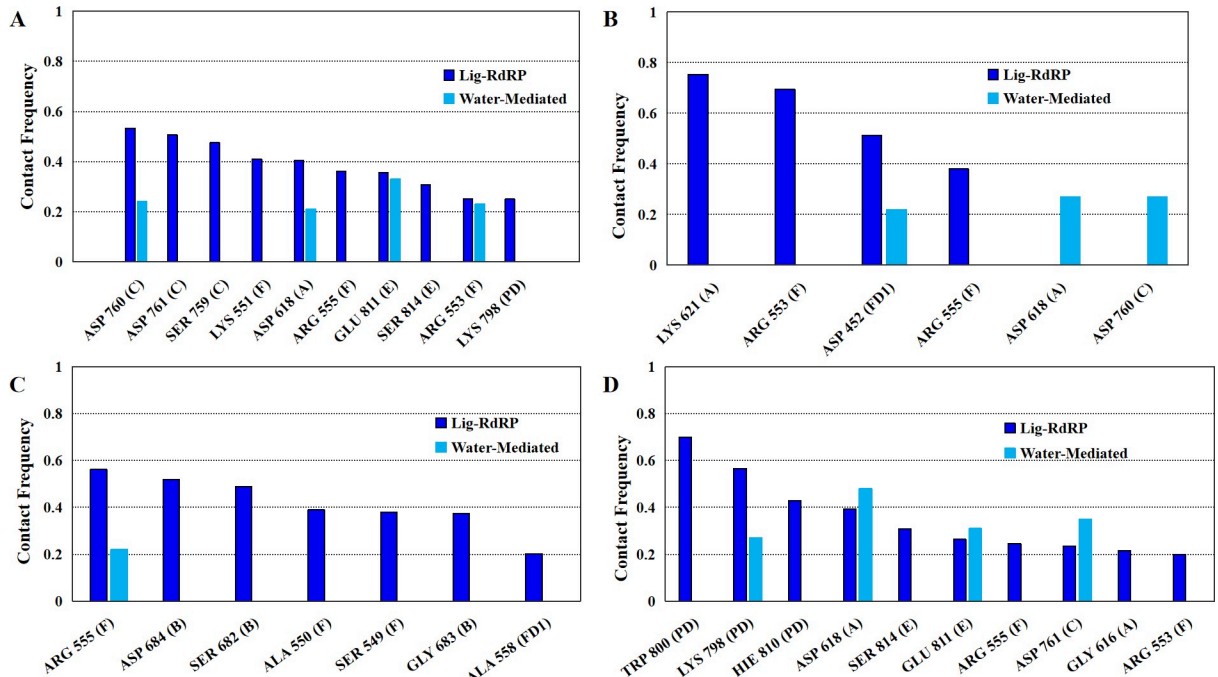

**Fig 4. Stable interactions formed by the phytochemicals.** Contact frequency of the hydrogen-bond and water mediated interactions between the RdRP residues and (A) SWE, (B) AMR, (C) SIT and (D) COR.

Residues from the conserved motifs A, C, E and F were observed to be forming interactions with SWE (Fig 4A). The three residues of conserved motif C were SER 759, ASP 760, and ASP 761 which are known to form one of the catalytic sites (SDD) of RdRP. ASP 760 was observed to be involved in water mediated interactions also. ASP 618 from the motif A was observed to strongly interact with SWE through direct and water mediated hydrogen bonding. GLU 811 and SER 814 from the motif E were involved in interacting with SWE with the former one forming water mediated interactions too. LYS 551, ARG 553 and ARG 555 from the ligand binding motif F interacted with SWE, ARG 553 was also involved in the water mediated hydrogen bonding. Among all these interacting residues SER 759, ASP 760, ASP 761, SER 814 are known to be involved in primer binding [31]. LYS 798 from the palm domain was also observed to interact with SWE.

Fig 4B depicts the interactions formed by AMR. LYS 621 from motif A was observed to form the strongest interaction. This was followed by ARG 553, ASP 452 and ARG 555. ARG 553 and ARG 555 belong to the ligand binding motif F. ASP 452 which belongs to the FD1 subdomain was involved in formation of water mediated interactions. ASP 618 and ASP 760 from motifs A and C respectively, were observed to be involved in forming strong water mediated hydrogen bonds. Among all these residues ASP 760 is part of the catalytic site SDD of motif C and is also known to be involved in primer binding.

Fig 4C shows the interaction between SIT and the residues of RdRP. The residues involved were ARG 555, ASP 684, SER 682, ALA 550, SER 549, GLY 683 and ALA 558. ARG 555 was also involved in formation of water mediated interactions with SIT. SER 682, GLY 683 and ASP 684 from the motif B are known to be involved in binding to the RNA template. SER 549 and ALA 550 belong to the motif F and known to be involved in inhibitor binding. ALA 558 is part of the FD1 subdomain and plays a role in binding to the template.

Fig 4D shows the interactions formed between COR and the residues of RdRP. TRP 800, LYS 798 and HIE 810 from the PD subdomain were observed to strongly interact with COR. LYS 798 was also involved in formation of water mediated interactions. HIE 810 is one of the crucial residues of the catalytic site. GLY 616 and ASP 618 belonged to motif A. ASP 761 was observed to form water mediated interactions, and is known to be involved in primer binding and also a part of the catalytic residues. GLU 811 and SER 814 are from the motif E, GLU 811 formed water mediated interactions. SER 814 is also known to be involved in primer binding. ARG 553 and ARG 555 from the ligand binding motif F were involved in forming strong interactions with COR.

The aspartate and serine residues from the catalytic triads SDD and CSQ, respectively, were observed to form non-bonded interactions with either of the four phytochemicals. Both of these catalytic triads are also known to be involved in primer binding [31]. One of the interesting findings was involvement of stable water assisted interactions between the phytochemicals and RdRP residues from the catalytic sites. ARG 555 from motif F which is known to interact with RdRP inhibitors was observed to interact with all the four phytochemicals. SIT and COR were observed to interact with residues from the PD1 subdomain which are known to be important in binding to the RNA template. The ranking of the four phytochemicals in decreasing order of number of interactions formed was COR and SWE followed by SIT, and AMR.

## Binding efficiency of phytochemicals

The thermodynamic estimation of the interactions between the phytochemicals and RdRP residues was done by calculating the MM-GBSA free energy of binding between the two [61]. S4 Fig in S1 File shows the histogram plot for the distribution of conformers along the free energy of binding between the phytochemical and the RdRP protein. All the phytochemicals except

for AMR showed free energy values ranging within -10 to -50 kcal/mol. Based on this distribution of histograms an optimal average of the free energy of binding (Optimal$^{Avg}$FE) was calculated using the equation given below,

$$\text{Optimal}^{\text{Avg}}\,\text{FE} = \frac{\sum_{i=1}^{n} E_i P_i}{P_T}$$

Where, $n$ is the total number of bins calculated, $E_i$ is the free energy value at which the bin $i$ is calculated, $P_i$ is the size of the population for the bin $i$ and $P_T$ is the size of the total population under consideration.

Fig 5 shows the Optimal$^{Avg}$FE values between the phytochemicals SWE (black), AMR (red), SIT (green), COR (blue) and the RdRP protein. It was observed that SIT showed the best value for Optimal$^{Avg}$FE followed by COR, SWE and AMR. However, there was a difference of around 1.4 kcal/mol between SIT and COR. This difference was considerably small as compared to the difference of 7.8 and 8.8 kcal/mol between SIT and SWE, SIT and AMR, respectively. These free energy values suggests that SIT and COR showed comparatively better binding than SWE and AMR.

The residue-wise contribution in free energy of binding between the phytochemicals and RdRP was calculated for 26 residues (Fig 6). PLIP software was used to find out the residues that were involved in either hydrophilic or hydrophobic interactions with the phytochemicals [65]. It was observed that a total of twenty-six residues namely, ASP 452, TYR 455, LYS 500, SER 549, ALA 550, ALA 553, ARG 555, GLY 616, ASP 618, LYS 621, CYS 622, ASP 623, ARG 624, SER 682, GLY 683, ASP 684, SER 759, ASP 760, ASP 761, LYS 798, TRP 800, GLU 811, CYS 813, SER 814 and ARG 836, were common to all the four phytochemicals. Most of these residues belonged to either of the seven conserved motifs. Five of the six residues from the two catalytic sites SDD (SER 759, ASP 760, ASP 761) from motif C and CSQ (CYS 813, SER 814, GLN 815) from motif E were observed to be involved in forming interactions with the phytochemicals. SER 549, ALA 550, ALA 553 and ARG 555 belonged to the ligand-binding motif F. LYS 500, SER 682, GLY 683, and ASP 684 are known to bind to the RNA template and are a part of the FD1 and FD2 subdomains and also belong to motifs G and B. GLY 616, ASP 618, LYS 621, CYS 622, ASP 623, and ARG 624 belong to motif A. ASP 452 (FD1), TYR 455 (FD1), LYS 798 (PD), TRP 800 (PD) and ARG 836 (TD) do not belong to any conserved motif but are a part of the three crucial subdomains. Fig 6A–6D shows the free energy contribution of these residues in binding to SWE, AMR, SIT and COR, respectively. A free energy value lower than

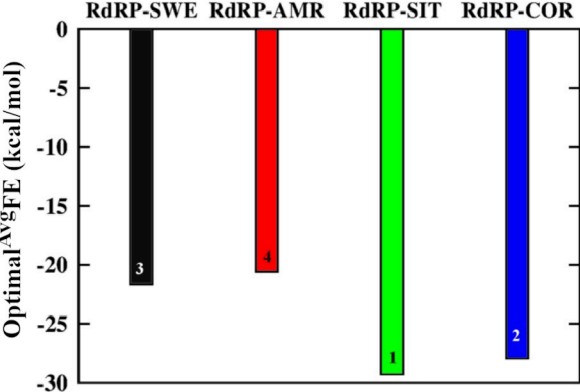

**Fig 5. Free energy of binding of phytochemicals.** Optimal average of free energy of binding between the phytochemicals SWE (black), AMR (red), SIT (green) and COR (blue) and RdRP protein.

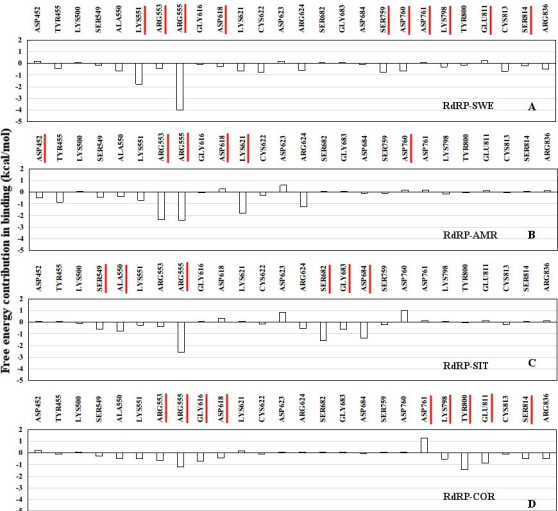

**Fig 6. Contribution in binding by interacting residues.** Residue-wise free energy contribution in binding between the ligands (A) SWE, (B) AMR, (C) SIT, and (D) COR and the RdRP protein.

zero indicates favorable binding. The two arginine residues from the ligand binding motif F, namely ARG 553 and ARG 555 were observed to show significantly better free energy of binding with all the phytochemicals in comparison to the rest of the residues. Most of the residues that were observed to form hydrogen bond or salt bridge interactions (Fig 4) with either of the four phytochemicals had considerably better free energy values. The best free energy contribution was by ARG 555 in binding to the SWE in comparison to all others. Comparatively, more number of residues were observed to bind favorably in case of SIT and COR than SWE and AMR. SER 682, GLY 683 and ARG 684 which belong to motif B and are known to be involved in binding to the RNA template showed favorable binding to SIT. Similarly, LYS 798, TRP 800 and GLU 811 from the PD subdomain and motif E showed the most favorable binding to the COR. Apart from ARG 553 and ARG 555, SER 549, ALA 550, LYS 551 from the ligand binding motif F were observed to be binding favorably to all the four phytochemicals.

## Predominant RdRP residues in inhibitor binding

One of the earlier simulation studies reported by the authors had described the interacting residues of RdRP with nucleotide analogues [51]. Six nucleotide analogues were studied, namely, favipiravir, galidesivir, lamivudine, ribavirin, remdesivir and sofosbuvir. Fig 7 lists the common RdRP residues that were observed to be involved in interacting with the phytochemicals and these nucleoside analogues. It was observed that overall twenty-six and twenty RdRP residues were involved in binding to the phytochemicals and nucleoside analogues respectively. Among these, seventeen residues were observed to be common to both. The function-site interaction fingerprint method performed by Zhoa Z and Bourne PE, suggests the interaction with inhibitors by LYS 551, ARG 553, ASP 623, SER 682 and ASP 760 residues from the different conserved motifs of RdRP [30]. Similar interactions were observed in the experimental structure of remdesivir-bound RNA in complex with RdRP of SARS-CoV-2 [31]. In the present study, these six residues were obtained to be interacting with all the four phytochemicals. The free energy values also showed that these six residues showed favorable binding (Fig 6). ASP 760 which is a catalytic site residue was involved in formation of water mediated interactions with SWE and AMR. SER 682 formed hydrogen bonds and salt bridges with SIT. COR

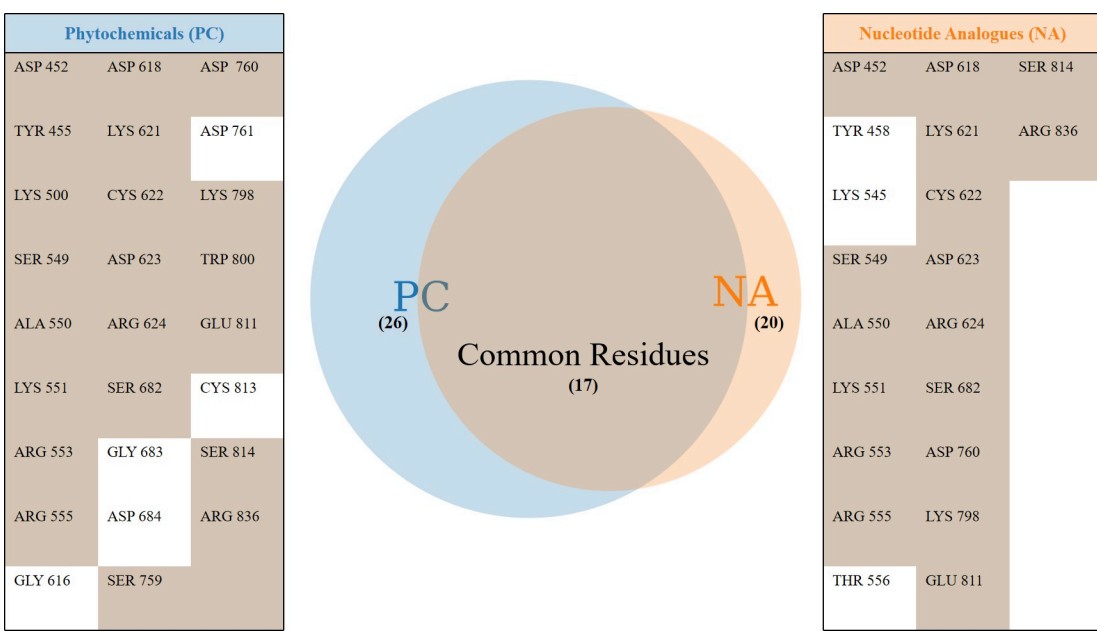

**Fig 7. Inhibitor binding RdRP residues.** Unique and common RdRP residues interacting with phytochemicals (PC) and nucleotide analogues (NA) observed through MD simulations.

formed hydrogen bonds with ARG 553. Although, COR showed no significant interaction with ASP 760, it formed strong water-mediated hydrogen bond with its neighbouring catalytic site residue ASP 761. This suggests the importance of the two aspartate residues which have also been reported earlier by Aftab et al. [20]. Similarly, one of the earlier reported computational studies on remdesivir binding to RdRP suggests the role of SER 549, ARG 555, ASP 618 and LYS 798 [27]. ARG 555 and ASP 618 were observed to form stable hydrogen bond/salt bridge interactions with SWE, AMR and COR. However, SER 549 was observed to interact more with SIT as compared to the other three phytochemicals (Figs 4C and 6). LYS 798 was observed to form hydrogen bonding with SWE and COR. Virtual screening studies on small molecules that are non-nucleotide inhibitors, as are the phytochemicals, suggests the role of GLU 811 and the CSQ catalytic site in binding to the inhibitors [29]. SWE and COR were observed to form water mediated interactions with GLU 811 (Fig 4A and 4D). SER 814 from the CSQ catalytic site was also observed to form hydrogen bond/salt bridge interactions with SWE and COR. These observations suggest that the important RdRP residues known to be involved in its inhibition showed significant interactions with the phytochemicals.

## Interacting moieties of phytochemicals

The four phytochemicals explored in this study consist of a withanolide derivative and glycosides. SWE is a xanthone glycoside known to have anti-viral properties [66]. AMR is a bitter secoiridoids glycoside, which possess anti-microbial properties [67]. SIT is a glycowithanolide known to have antineoplastic activity [68]. COR is a sulfur-containing clerodane diterpene glycoside, with known immunomodulatory properties [69]. The glucose moiety appeared to be the common functional group in each of these four phytochemcials. This moiety was flanked by either hydrophilic or hydrophobic functional groups. Fig 8 shows the snapshot of RdRP residues interacting with the phytochemicals. The glucose moieties have been shown in red circle. SWE has a hydroxyl-methoxyxanthone ring which was observed to interact with the postively

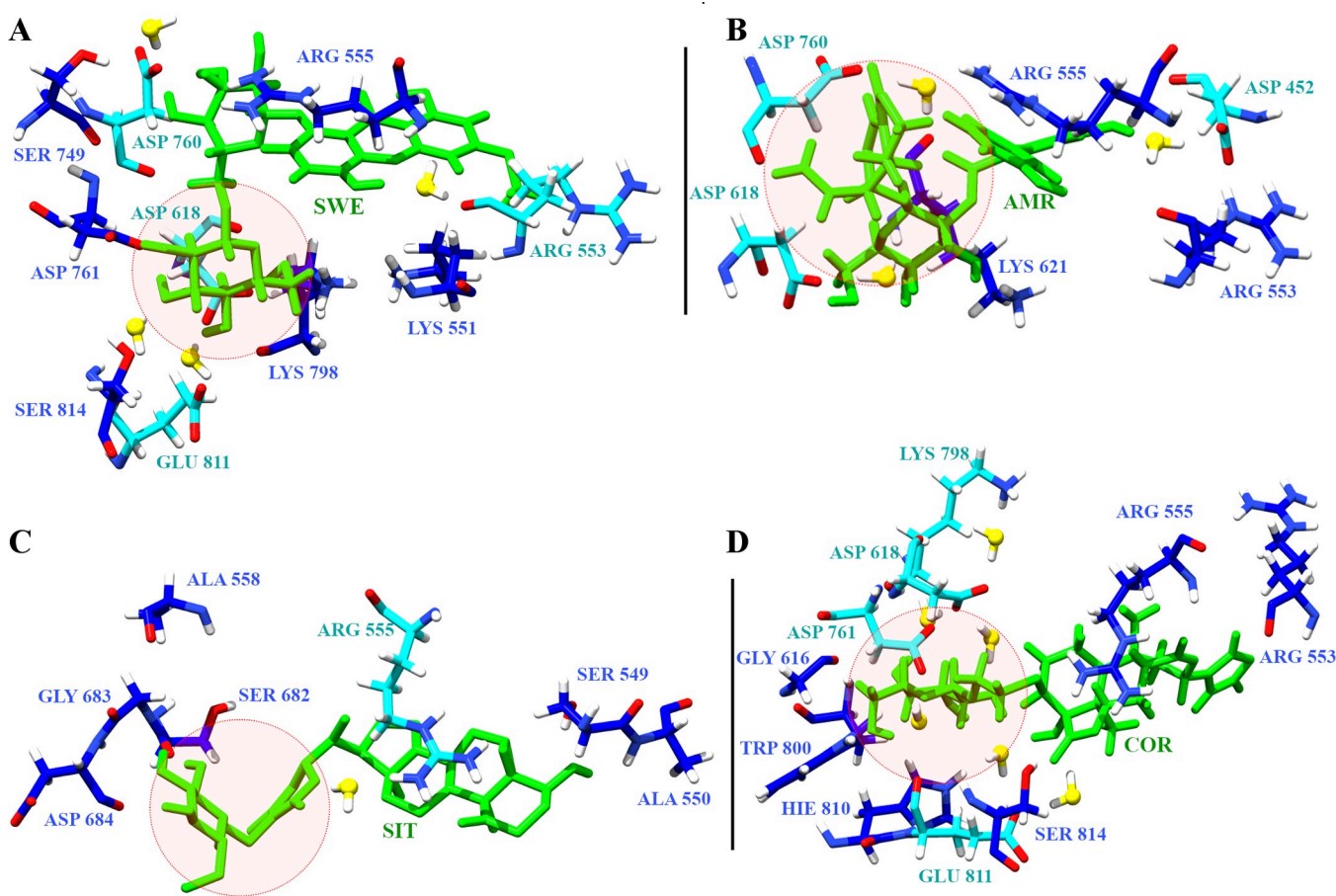

**Fig 8. Phytochemical interaction partners.** Snapshots of RdRP residues forming hydrogen bonds (blue) and water mediated (cyan) interactions with (A) SWE, (B) AMR, (C) SIT and (D) COR. The red circle denotes the interacting glucose moieties of the phytochemicals.

charged amino acid arginine (Fig 8A). The other two glucopyranoside rings were observed to form strong water mediated interactions with negatively charged amino acids aspartate and glutamate. AMR has an iridoid and a trihydroxy-biphenyl-carboxylic acid moiety [67]. They were observed to form stable interactions with aspartate residues (Fig 8B). The glucose moieties in SIT and COR were observed to form water mediated interactions with charged amino acids, namely, arginine, lysine and aspartate (Fig 8C and 8D) [68, 69]. The free energy of binding values for these two phytochemicals suggests that the interaction of charged amino acids with the glucose moieties may be energetically more favourable. Hence, it may be inferred that these glycoside derivatives prove to be efficient binders when their interacting partners are hydrophilic and charged amino acids.

## Conclusion

The present study explains the capability of phytochemicals from Indian medicinal plants in forming stable interactions with the RdRP of the SARS-CoV-2. Previously, decoded mechanisms of pre-mature RNA termination by drugs indicate the role of dominant molecular motions in the finger and thumb subdomains of the RdRP. Similarly, the molecular dynamics simulations reported here, comprehend these molecular motions through PCA and distance calculations. The phytochemical-bound RdRP systems sampled closed conformations of the

template entry site. Whereas, the ligand-free RdRP system pre-dominantly showed only open conformation for the same. Hydrogen bonding, salt bridge interactions and free energy calculations suggest favourable binding of sitoindoside IX, cordifolide A and swertiapuniside. However, cordifolide A with minimal difference in free energy of binding against sitoindoside IX was observed to form long lasting water mediated hydrogen bonds with the catalytic site aspartate residues. It may infer that it has more potential in strong binding to RdRP as compared to the remaining three phytochemicals. The key approach in designing a small molecular inhibitor lies in their prospects to form energetically favourable interactions within the drug-target. ARG 553, ARG 555, ASP 618, ASP 760, ASP 761, GLU 811, SER 814 were observed to participate in such interactions. Here, the interacting partners were the glycosidic moieties of these phytochemicals. Overall, the finding reported in this article proposes the phytochemicals as stable RdRP catalytic site binders, synergistic with the experimentally known drug-RdRP interactions.

## Supporting information

**S1 File.**
(DOCX)

## Acknowledgments

The authors would also like to thank the National PARAM Supercomputing Facility (NPSF) and Bioinformatics Resources and Applications Facility (BRAF) for the computing infrastructure

## Author Contributions

**Conceptualization:** Shruti Koulgi, Vinod Jani, Uddhavesh Sonavane, Rajendra Joshi.

**Data curation:** Shruti Koulgi, Vinod Jani, Mallikarjunachari Uppuladinne V. N.

**Formal analysis:** Shruti Koulgi, Vinod Jani, Mallikarjunachari Uppuladinne V. N.

**Investigation:** Shruti Koulgi, Vinod Jani, Mallikarjunachari Uppuladinne V. N.

**Methodology:** Shruti Koulgi, Vinod Jani, Mallikarjunachari Uppuladinne V. N., Uddhavesh Sonavane.

**Supervision:** Rajendra Joshi.

**Visualization:** Shruti Koulgi, Vinod Jani, Mallikarjunachari Uppuladinne V. N.

**Writing – original draft:** Shruti Koulgi, Vinod Jani, Mallikarjunachari Uppuladinne V. N.

**Writing – review & editing:** Shruti Koulgi, Vinod Jani, Mallikarjunachari Uppuladinne V. N., Uddhavesh Sonavane, Rajendra Joshi.

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
