## [Decision Letter · Decision Letter 0]

30 Mar 2021

PONE-D-20-38144

Molecular insights into the inhibitory effect of phytochemicals on RNA dependent RNA polymerase of Severe Acute Respiratory Syndrome Coronavirus-2

PLOS ONE

Dear Dr. Joshi,

Thank you for submitting your manuscript to PLOS ONE. After careful consideration, we feel that it has merit but does not fully meet PLOS ONE’s publication criteria as it currently stands. Therefore, we invite you to submit a revised version of the manuscript that addresses the points raised during the review process.

We look forward to receiving your revised manuscript.

Kind regards,

Chandrabose Selvaraj, Ph.D.

Academic Editor

PLOS ONE

Journal Requirements:

[The authors would like to acknowledge the National Supercomputing Mission for providing financial support for this work.]

 [The funders had no role in study design, data collection and analysis, decision to publish, or preparation of the manuscript.]

Reviewers' comments:

Reviewer's Responses to Questions

5. Review Comments to the Author

Reviewer #1: Authors performed in silico analyses to study the molecular analyses and the inhibitory effect of phytochemicals on RNA dependent RNA polymerase of Severe Acute Respiratory Syndrome Coronavirus-2. The article is interested however following points will help to improve the article.

1- The title should be more attractive.

2- Abstract has some confused statements. Please rewrite.

3- Discussion section is poorly drafted.

4- There are numerous typos and linguistics errors.

5- Figures are poorly drafted. Please improve.

6- Conclusion section is too long and a big chunk seems to be copied from Abstract. Please re write

Reviewer #2: The problem with computational methods is that they always show results, even when the starting points are useless. This is the reason why one should be really careful about the setup of such computational methods. The current study mixed several approaches to earn the publication, however, they failed to enhance the more consistent general research approach or a more obvious integration of computational techniques, to reach to a suitable conclusion. A validation of their computational approach is needed in this research. Ultimately, I do not think that the manuscript is suitable to process further.

Reviewer #3: 1) Overall the language needs to be corrected as I could see non-scientific way of explaining all through out the manuscript. For example "The availability of immense structural knowledge on one of the hottest drug target of SARS-CoV-2,...."

2) Also the manuscript requires significant re-writing as the sentences go on like a paragraph and hard to get along with the flow of the manuscript.

3) The introduction is exhaustive and should be reduced to maximum of two pages with key elements of the study.

4) I am not sure why the authors have discussed few results and methodology in the introduction section.

5) The authors mention that "The availability of immense structural knowledge on one of the hottest drug target of SARS-CoV-2, RdRP and in-silico techniques like, molecular docking and simulations would enable the designing of inhibitors based on

screening of phytochemicals from medicinal plants specific to treating respiratory ailments" these phytochemicals from 12 medicinal plants evaluated in this current study were proven to treat respiratory ailments. But the authors fail to cite the references. So it is advisable to include the references for all 12 medicinal plants showing their potential relevance against respiratory diseases.

6) Were the phytochemicals from these 12 medicinal plants reported after GC-MS analysis, if so cite the refence for that.

7) The authors need to mention, how many phytochemicals were extracted out of these 12 medicinal plants ?

8) Did the authors check ADMET property of these phytochemicals ? if so furnish the ADMET data.

9) Bond length information should be listed as table for at least top 10 interactions.

10) Discussion has to be more directed towards the chemical nature of the compound and it's interacting partners, this will be more interesting rather than focusing on the amino acid residues at the active site. Hence, the authors are encouraged to focus more on the chemical entities of the phytochemicals and discuss about it's medicinal property.

---

## [Author Response · Author response to Decision Letter 0]

26 Apr 2021

The authors would like to thank the editor and the reviewers for their valuable suggestions and comments. They have helped in enhancing the scientific merit and readability of the manuscript. The authors sincerely hope that the revised manuscript would satisfy the concerns raised by the reviewers. A detailed response to every comment/suggestion has been given (highlighted in yellow). The additions made in the revised manuscript have been highlighted in grey.

Reviewer #1: Authors performed in silico analyses to study the molecular analyses and the inhibitory effect of phytochemicals on RNA dependent RNA polymerase of Severe Acute Respiratory Syndrome Coronavirus-2. The article is interested however following points will help to improve the article.

1-The title should be more attractive.

Response: The authors have modified the title as per reviewer’s suggestion. The new title has been given below.

Natural plant products as potential inhibitors of RNA dependent RNA polymerase of Severe Acute Respiratory Syndrome Coronavirus-2

2-Abstract has some confused statements. Please rewrite.

Response: The abstract has been re-written to avoid any confused statements. The changes made in the abstract of the revised manuscript have been highlighted.

3-Discussion section is poorly drafted.

Response: The discussion section has been revisited and re-drafted for better understanding of the results. The changes made have been highlighted in yellow in the revised manuscript.

4-There are numerous typos and linguistics errors.

Response: The entire manuscript has been revised to rectify the grammatical errors.

5-Figures are poorly drafted. Please improve.

Response: As per the reviewer’s suggestion, few of the figures have been redrafted for better understanding of the results. The old figures 1, 4, 5, 6, and 7 have been modified in the revised manuscript. The revised manuscript now contains eight figures. The 2D-representations of the four phytochemicals in the old figure 1 has now been shifted to supplementary figure S2. The revised figure 1 has been given below.

Fig 1. Structure of RdRP of SARS-CoV-2. The different (A) subdomains and (B) the seven conserved motifs in RdRP of SARS-CoV-2 represented through color codes.

The sub-figure A of the old figures 4-7 have been removed as they were observed to be repetitive with the sub-figure B. Revised figure 4 now shows the contact frequency of the hydrogen bond and water mediated interactions for (A) SWE, (B) AMR, (C) SIT and (D) COR. The revised figure 4 has been given below.

Fig 4. Stable interactions formed by the phytochemicals. Contact frequency of the hydrogen-bond and water mediated interactions between the RdRP residues and (A) SWE, (B) AMR, (C) SIT and (D) COR.

The sub-figure C of the old figures 4-7 have been included as a new Figure 8 in the revised manuscript. The new figure 8 talks about the functional moieties of the phytochemicals interacting with different amino acids of RdRP. The new figure 8 from the revised manuscript has been given below. A paragraph describing this figure has also been included (as per one of the reviewer’s suggestions).

Fig 8. Phytochemical interaction partners. Snapshots of RdRP residues forming hydrogen bonds (blue) and water mediated (cyan) interactions with (A) SWE, (B) AMR, (C) SIT and (D) COR. The red circle denotes the interacting glucose moieties of the phytochemicals.

The following paragraph has been included in the “Results and Discussion”

Interacting moieties of phytochemicals

The four phytochemicals explored in this study consist of a withanolide derivative and glycosides. SWE is a xanthone glycoside known to have anti-viral properties [66]. AMR is a bitter secoiridoids glycoside, which possess anti-microbial properties [67]. SIT is a glycowithanolide known to have antineoplastic activity [68]. COR is a sulfur-containing clerodane diterpene glycoside, with known immunomodulatory properties [69]. The glucose moiety appeared to be the common functional group in each of these four phytochemcials. This moiety was flanked by either hydrophilic or hydrophobic functional groups. Fig 8 shows the snapshot of RdRP residues interacting with the phytochemicals. The glucose moieties have been shown in red circle. SWE has a hydroxyl-methoxyxanthone ring which was observed to interact with the postively charged amino acid arginine (Fig 8 A). The other two glucopyranoside rings were observed to form strong water mediated interactions with negatively charged amino acids aspartate and glutamate. AMR has an iridoid and a trihydroxy-biphenyl-carboxylic acid moiety [67]. They were observed to form stable interactions with aspartate residues (Fig 8 B). The glucose moieties in SIT and COR were observed to form water mediated interactions with charged amino acids, namely, arginine, lysine and aspartate (Fig 8 C and D) [68, 69]. The free energy of binding values for these two phytochemicals suggests that the interaction of charged amino acids with the glucose moieties may be energetically more favourable. Hence, it may be inferred that these glycoside derivatives prove to be efficient binders when their interacting partners are hydrophilic and charged amino acids.

6-Conclusion section is too long and a big chunk seems to be copied from Abstract. Please re write

Response: The “Conclusion” section in the revised manuscript has been re-drafted and there are no repetitions.

Reviewer #2: The problem with computational methods is that they always show results, even when the starting points are useless. This is the reason why one should be really careful about the setup of such computational methods. The current study mixed several approaches to earn the publication, however, they failed to enhance the more consistent general research approach or a more obvious integration of computational techniques, to reach to a suitable conclusion. A validation of their computational approach is needed in this research. Ultimately, I do not think that the manuscript is suitable to process further.

Response: The authors agree to the fact that the results need the validation through experiments. However, currently the observations have been supported by the few of the literature reports which have used similar computational approaches. 

Reviewer #3: 

1)Overall the language needs to be corrected as I could see non-scientific way of explaining all through out the manuscript. For example "The availability of immense structural knowledge on one of the hottest drug target of SARS-CoV-2,...."

Response: The manuscript has been revisited and revised according to the reviewer’s suggestions.

2)Also the manuscript requires significant re-writing as the sentences go on like a paragraph and hard to get along with the flow of the manuscript.

Response: The manuscript has been re-written and the sentences have been shortened or split in order to improve the readability.

3)The introduction is exhaustive and should be reduced to maximum of two pages with key elements of the study.

Response: The “Introduction” section has been reduced considerably in the revised manuscript.

4)I am not sure why the authors have discussed few results and methodology in the introduction section.

Response: The “Introduction” section has been reduced considerably in the revised manuscript. The sentences explaining the results and methodology have now been removed from the “Introduction” of the revised manuscript.

5) The authors mention that "The availability of immense structural knowledge on one of the hottest drug target of SARS-CoV-2, RdRP and in-silico techniques like, molecular docking and simulations would enable the designing of inhibitors based on

screening of phytochemicals from medicinal plants specific to treating respiratory ailments" these phytochemicals from 12 medicinal plants evaluated in this current study were proven to treat respiratory ailments. But the authors fail to cite the references. So it is advisable to include the references for all 12 medicinal plants showing their potential relevance against respiratory diseases.

Response: The references for all twelve medicinal plants included in this study have been added and have also been cited accordingly. The references have been mentioned along with the supplementary Figure S1. The revised supplementary figure S1 has been given below.

S1 Fig. Methodology. Protocol followed for creation of the phytochemical dataset with the list of the Indian medicinal plant sources.

References numbered as superscripts beside the scientific name of the plants:

1.Saini A, Sharma S, Chhibber S. Induction of resistance to respiratory tract infection with Klebsiella pneumoniae in mice fed on a diet supplemented with tulsi (Ocimum sanctum) and clove (Syzgium aromaticum) oils. J Microbiol Immunol Infect. 2009 Apr;42(2):107-13. PMID: 19597641.

2.Cohen MM. Tulsi - Ocimum sanctum: A herb for all reasons. J Ayurveda Integr Med. 2014 Oct-Dec;5(4):251-9. doi: 10.4103/0975-9476.146554. PMID: 25624701; PMCID: PMC4296439.

3.Lee JW, Ryu HW, Park SY, Park HA, Kwon OK, Yuk HJ, Shrestha KK, Park M, Kim JH, Lee S, Oh SR, Ahn KS. Protective effects of neem (Azadirachta indica A. Juss.) leaf extract against cigarette smoke- and lipopolysaccharide-induced pulmonary inflammation. Int J Mol Med. 2017 Dec;40(6):1932-1940. doi: 10.3892/ijmm.2017.3178. Epub 2017 Oct 10. PMID: 29039495.

4.Jose Francisco Islas, Ezeiza Acosta, Zuca G-Buentello, Juan Luis Delgado-Gallegos, María Guadalupe Moreno-Treviño, Bruno Escalante, Jorge E. Moreno-Cuevas. An overview of Neem (Azadirachta indica) and its potential impact on health. Journal of Functional Foods, Volume 74, 2020, 104171, ISSN 1756-4646,doi: 10.1016/j.jff.2020.104171.\\

5.Kumari M, Ashok BK, Ravishankar B, Pandya TN, Acharya R. Anti-inflammatory activity of two varieties of Pippali (Piper longum Linn.). Ayu. 2012 Apr;33(2):307-10. doi: 10.4103/0974-8520.105258. PMID: 23559810; PMCID: PMC3611634.

6.Saha S, Ghosh S. Tinospora cordifolia: One plant, many roles. Anc Sci Life. 2012 Apr;31(4):151-9. doi: 10.4103/0257-7941.107344. PMID: 23661861; PMCID: PMC3644751.

7.Sharma P, Dwivedee BP, Bisht D, Dash AK, Kumar D. The chemical constituents and diverse pharmacological importance of Tinospora cordifolia. Heliyon. 2019 Sep 12;5(9):e02437. doi: 10.1016/j.heliyon.2019.e02437. PMID: 31701036; PMCID: PMC6827274.

8.Antul, K., P. Amandeep, S. Gurwinder, and C. Anuj. “Review on Pharmacological Profile of Medicinal Vine: Tinospora Cordifolia”. Current Journal of Applied Science and Technology, Vol. 35, no. 5, June 2019, pp. 1-11, doi:10.9734/cjast/2019/v35i530196.

9.Kumar V, Van Staden J. A Review of Swertia chirayita (Gentianaceae) as a Traditional Medicinal Plant. Front Pharmacol. 2016 Jan 12;6:308. doi: 10.3389/fphar.2015.00308. PMID: 26793105; PMCID: PMC4709473.

10.Dey P, Singh J, Suluvoy JK, Dilip KJ, Nayak J. Utilization of Swertia chirayita Plant Extracts for Management of Diabetes and Associated Disorders: Present Status, Future Prospects and Limitations. Nat Prod Bioprospect. 2020 Dec;10(6):431-443. doi: 10.1007/s13659-020-00277-7. Epub 2020 Oct 28. PMID: 33118125; PMCID: PMC7648839.

11.Koul A, Bala S, Yasmeen, Arora N. Aloe vera affects changes induced in pulmonary tissue of mice caused by cigarette smoke inhalation. Environ Toxicol. 2015 Sep;30(9):999-1013. doi: 10.1002/tox.21973. Epub 2014 Feb 24. PMID: 24615921.

12.Zayas LE, Wisniewski AM, Cadzow RB, Tumiel-Berhalter LM. Knowledge and use of ethnomedical treatments for asthma among Puerto Ricans in an urban community. Ann Fam Med. 2011 Jan-Feb;9(1):50-6. doi: 10.1370/afm.1200. PMID: 21242561; PMCID: PMC3022046.

13.Shridhar Dwivedi, Terminalia arjuna Wight & Arn.—A useful drug for cardiovascular disorders, Journal of Ethnopharmacology, Volume 114, Issue 2, 2007, Pages 114-129, ISSN 0378-8741, https://doi.org/10.1016/j.jep.2007.08.003.

14.Townsend EA, Siviski ME, Zhang Y, Xu C, Hoonjan B, Emala CW. Effects of ginger and its constituents on airway smooth muscle relaxation and calcium regulation. Am J Respir Cell Mol Biol. 2013 Feb;48(2):157-63. doi: 10.1165/rcmb.2012-0231OC. Epub 2012 Oct 11. PMID: 23065130; PMCID: PMC3604064.

15.Anh NH, Kim SJ, Long NP, Min JE, Yoon YC, Lee EG, Kim M, Kim TJ, Yang YY, Son EY, Yoon SJ, Diem NC, Kim HM, Kwon SW. Ginger on Human Health: A Comprehensive Systematic Review of 109 Randomized Controlled Trials. Nutrients. 2020 Jan 6;12(1):157. doi: 10.3390/nu12010157. PMID: 31935866; PMCID: PMC7019938.

16.Choudhary B, Shetty A, Langade DG. Efficacy of Ashwagandha (Withania somnifera [L.] Dunal) in improving cardiorespiratory endurance in healthy athletic adults. Ayu. 2015 Jan-Mar;36(1):63-8. doi: 10.4103/0974-8520.169002. PMID: 26730141; PMCID: PMC4687242.

17.Rahmani AH, Alsahli MA, Aly SM, Khan MA, Aldebasi YH. Role of Curcumin in Disease Prevention and Treatment. Adv Biomed Res. 2018 Feb 28;7:38. doi: 10.4103/abr.abr_147_16. PMID: 29629341; PMCID: PMC5852989.

18.Samareh Fekri M, Poursalehi HR, Sharififar F, Mandegary A, Rostamzadeh F, Mahmoodi R. The effects of methanolic extract of Glycyrrhiza glabra on the prevention and treatment of bleomycin-induced pulmonary fibrosis in rat: experimental study. Drug Chem Toxicol. 2019 May 9:1-7. doi: 10.1080/01480545.2019.1606232. Epub ahead of print. PMID: 31072167.

19.Wang L, Yang R, Yuan B, Liu Y, Liu C. The antiviral and antimicrobial activities of licorice, a widely-used Chinese herb. Acta Pharm Sin B. 2015 Jul;5(4):310-5. doi: 10.1016/j.apsb.2015.05.005. Epub 2015 Jun 17. PMID: 26579460; PMCID: PMC4629407.

20.Okhuarobo A, Falodun JE, Erharuyi O, Imieje V, Falodun A, Langer P. Harnessing the medicinal properties of Andrographis paniculata for diseases and beyond: a review of its phytochemistry and pharmacology. Asian Pac J Trop Dis. 2014 Jun;4(3):213–22. doi: 10.1016/S2222-1808(14)60509-0. PMCID: PMC4032030.

5)Were the phytochemicals from these 12 medicinal plants reported after GC-MS analysis, if so cite the refence for that.

Response: These phytochemicals were selected based on their availability in PubChem and CAS databases. The PubChem CIDs and CAS IDs have been referred accordingly. No experiments were performed for selection of these phytochemicals. 

6)The authors need to mention, how many phytochemicals were extracted out of these 12 medicinal plants ?

Response: A total of 150 phytochemicals were selected out of the 12 medicinal plants. This information has been included in the revised manuscript.

7)Did the authors check ADMET property of these phytochemicals ? if so furnish the ADMET data.

Response: The authors have checked the ADMET property for the top-ranked four phytochemicals, namely, swertiapuniside, cordifolide A, sitoindoside IX and amarogentin. A supplementary table S1 with a short paragraph explaining the ADMET properties has now been added in the supplementary data. The supplementary table 1 along with the explanation included in the revised supplementary data has been given below.

S1 Table. ADMET properties. ADMET properties of the four phytochemicals.

    ABSORPTION DISTRIBUTION METABOLISM TOXICITY

Medicinal Plant (Source) Phytochemical Name GI WS SP

(Log Kp) BBB Crossing Subcellular Localization P-gly sub hERG inb CYP1A2 inb CYP2C19 inb 

A*CYP2D6 inb CYP3A4 inb CAR 

Swertia chirayita Swertiapuniside +  -1.64 -11.04 - Mitochondria + + - - - - -

 Amarogentin + -2.97 -8.16 - Mitochondria - - - - - - -

Withania somnifera Sitoindoside IX +  -3.80 -8.97 + Mitochondria + + - - - - -

Tinospora cordifolia Cordifolide A - -3.542 -9.7 - Mitochondria + + - - - - -

The gastrointestinal absorption (GI), water solubility (WS) and skin permeation (SP) explains the absorption properties. Cordifolide A was observed to show low GI absorption, whereas, the remaining three showed favourable values for the same. The other two parameters showed allowed values for all the four phytochemicals. The blood-brain-barrier (BBB) crossing, subcellular localization and P-glycoprotein substrate (P-gly sub) parameters denote the distribution properties. All the phytochemicals were predicted to have the subcellular localization in the mitochondria. Except for amarogentin, all were predicted to serve as the P-glycoprotein substrate. Human ether-a-go-go Related Gene (hERG), cytochrome 1A2 (CYP1A2), cytochrome 2C19 (CYP2C19), cytochrome

2D6 (CYP2D6) and cytochrome 3A4 (CYP3A4) inhibitory property was calculated to understand the metabolism of the phytochemicals.Neither of them were predicted to be CYP inhibitors nor possessing toxic properties.

8)Bond length information should be listed as table for at least top 10 interactions.

Response: The bond length information for the hydrogen bonds formed between the phytochemicals and the residues of RdRP has been now included as supplementary table S2. The supplementary table S2 has been given below.

S2 Table. Bond distance of hydrogen bonds. The average bond distance for the hydrogen bond formed between RdRP residues and SWE, AMR, SIT and COR

RdRP Residue Hydrogen Bond Length (Å)

Swertiapuniside (SWE)

ASP 760 (C) 2.99

ASP 761 (C) 2.84

SER 759 (C) 2.85

LYS 551 (F) 3.5

ASP 618 (A) 3.28

ARG 555 (F) 3.4

GLU 811 (E) 2.81

SER 814 (E) 2.98

ARG 553 (F) 2.65

LYS 798 (PD) 2.88

Amarogentin (AMR)

LYS 621 (A) 2.99

ARG 553 (F) 3.45

ASP 452 (FD1) 2.66

ARG 555 (F) 3.3

ASP 618 (A) 3.22

ASP 760 (C) 3.38

Sitoindoside IX (SIT)

ARG 555 (F) 2.82

ASP 684 (B) 2.63

SER 682 (B) 2.67

ALA 550 (F) 2.95

SER 549 (F) 2.71

GLY 683 (B) 2.8

ALA 558 (FD1) 3.13

Cordifolide A (COR)

TRP 800 (PD) 3.09

LYS 798 (PD) 2.42

HIE 810 (PD) 2.3

ASP 618 (A) 3.21

SER 814 (E) 2.84

GLU 811 (E) 2.99

ARG 555 (F) 3.4

ASP 761 (C) 2.79

GLY 616 (A) 2.82

ARG 553 (F) 3.17

9)Discussion has to be more directed towards the chemical nature of the compound and it's interacting partners, this will be more interesting rather than focusing on the amino acid residues at the active site. Hence, the authors are encouraged to focus more on the chemical entities of the phytochemicals and discuss about it's medicinal property.

Response: The authors have included a new sub-section “Interacting moieties of phytochemicals” in the “Results and Discussion”. This section talks about the chemical moieties of the phytochemicals that were observed to interact with the different residues of RdRP. The newly included sub-section is as below.

Interacting moieties of phytochemicals

The four phytochemicals explored in this study consist of a withanolide derivative and glycosides. SWE is a xanthone glycoside known to have anti-viral properties [66]. AMR is a bitter secoiridoids glycoside, which possess anti-microbial properties [67]. SIT is a glycowithanolide known to have antineoplastic activity [68]. COR is a sulfur-containing clerodane diterpene glycoside, with known immunomodulatory properties [69]. The glucose moiety appeared to be the common functional group in each of these four phytochemcials. This moiety was flanked by either hydrophilic or hydrophobic functional groups. Fig 8 shows the snapshot of RdRP residues interacting with the phytochemicals. The glucose moieties have been shown in red circle. SWE has a hydroxyl-methoxyxanthone ring which was observed to interact with the postively charged amino acid arginine (Fig 8 A). The other two glucopyranoside rings were observed to form strong water mediated interactions with negatively charged amino acids aspartate and glutamate. AMR has an iridoid and a trihydroxy-biphenyl-carboxylic acid moiety [67]. They were observed to form stable interactions with aspartate residues (Fig 8 B). The glucose moieties in SIT and COR were observed to form water mediated interactions with charged amino acids, namely, arginine, lysine and aspartate (Fig 8 C and D) [68, 69]. The free energy of binding values for these two phytochemicals suggests that the interaction of charged amino acids with the glucose moieties may be energetically more favourable. Hence, it may be inferred that these glycoside derivatives prove to be efficient binders when their interacting partners are hydrophilic and charged amino acids.

A new figure, Figure 8 depicting the interacting residues with the glucose moieties has been included in the revised manuscript. The new figure 8 has been given below.

Fig 8. Phytochemical interaction partners. Snapshots of RdRP residues forming hydrogen bonds (blue) and water mediated (cyan) interactions with (A) SWE, (B) AMR, (C) SIT and (D) COR. The red circle denotes the interacting glucose moieties of the phytochemicals.

---

## [Decision Letter · Decision Letter 1]

4 May 2021

Natural plant products as potential inhibitors of RNA dependent RNA polymerase of Severe Acute Respiratory Syndrome Coronavirus-2

PONE-D-20-38144R1

Dear Dr. Joshi,

We’re pleased to inform you that your manuscript has been judged scientifically suitable for publication and will be formally accepted for publication once it meets all outstanding technical requirements.

Kind regards,

Chandrabose Selvaraj, Ph.D.

Academic Editor

PLOS ONE

Additional Editor Comments (optional):

Reviewers' comments:

Reviewer's Responses to Questions

**Comments to the Author**

1. If the authors have adequately addressed your comments raised in a previous round of review and you feel that this manuscript is now acceptable for publication, you may indicate that here to bypass the “Comments to the Author” section, enter your conflict of interest statement in the “Confidential to Editor” section, and submit your "Accept" recommendation.

Reviewer #1: All comments have been addressed

Reviewer #2: All comments have been addressed

Reviewer #3: All comments have been addressed

2. Is the manuscript technically sound, and do the data support the conclusions?

Reviewer #1: Yes

Reviewer #2: Partly

Reviewer #3: Yes

3. Has the statistical analysis been performed appropriately and rigorously? 

Reviewer #1: N/A

Reviewer #2: N/A

Reviewer #3: Yes

4. Have the authors made all data underlying the findings in their manuscript fully available?

Reviewer #1: Yes

Reviewer #2: Yes

Reviewer #3: Yes

5. Is the manuscript presented in an intelligible fashion and written in standard English?

Reviewer #1: Yes

Reviewer #2: Yes

Reviewer #3: Yes

6. Review Comments to the Author

Reviewer #1: Accepted

nice efforts by the authors and also address all the questions effectively to meet the criteria of journal.

Reviewer #2: (No Response)

Reviewer #3: (No Response)

---

## [Editor Report · Acceptance letter]

6 May 2021

PONE-D-20-38144R1 

Natural plant products as potential inhibitors of RNA dependent RNA polymerase of Severe Acute Respiratory Syndrome Coronavirus-2 

Dear Dr. Joshi:

I'm pleased to inform you that your manuscript has been deemed suitable for publication in PLOS ONE. Congratulations! Your manuscript is now with our production department. 

Kind regards, 

on behalf of

Dr. Chandrabose Selvaraj 

Academic Editor

PLOS ONE